# Rule-Based Embedded HMMs Phoneme Classification to Improve Qur'anic Recitation Recognition

Ammar Mohammed Ali Alqadasi [1] , Mohd Shahrizal Sunar [2,3,*] , Sherzod Turaev [4,*] , Rawad Abdulghafor [1,*] , Md Sah Hj Salam [2], Abdulaziz Ali Saleh Alashbi [2], Ali Ahmed Salem [5] and Mohammed A. H. Ali [6]

[1] Department of Computer Science, Faculty of Information and Communication Technology, International Islamic University Malaysia, Kuala Lumpur 53100, Malaysia
[2] Faculty of Computing, Universiti Teknologi Malaysia, Johor Bahru 81310, Malaysia
[3] Media and Game Innovation Centre of Excellence, Institute of Human Centered Engineering, Universiti Teknologi Malaysia, Johor Bahru 81310, Malaysia
[4] Department of Computer Science and Software Engineering, College of Information Technology, United Arab Emirates University, Al Ain 15551, Abu Dhabi, United Arab Emirates
[5] Institute of High Voltage and High Current, School of Electrical Engineering, Universiti Teknologi Malaysia, Johor Bahru 81310, Malaysia
[6] Department of Mechanical Engineering, Faculty of Engineering, University of Malaya, Kuala Lumpur 50603, Malaysia
* Correspondence: shahrizal@utm.my (M.S.S.); sherzod@uaeu.ac.ae (S.T.); rawad@iium.edu.my (R.A.)

**Abstract:** Phoneme classification performance is a critical factor for the successful implementation of a speech recognition system. A mispronunciation of Arabic short vowels or long vowels can change the meaning of a complete sentence. However, correctly distinguishing phonemes with vowels in Quranic recitation (the Holy book of Muslims) is still a challenging problem even for state-of-the-art classification methods, where the duration of the phonemes is considered one of the important features in Quranic recitation, which is called Medd, which means that the phoneme lengthening is governed by strict rules. These features of recitation call for an additional classification of phonemes in Qur'anic recitation due to that the phonemes classification based on Arabic language characteristics is insufficient to recognize Tajweed rules, including the rules of Medd. This paper introduces a Rule-Based Phoneme Duration Algorithm to improve phoneme classification in Qur'anic recitation. The phonemes of the Qur'anic dataset contain 21 Ayats collected from 30 reciters and are carefully analyzed from a baseline HMM-based speech recognition model. Using the Hidden Markov Model with tied-state triphones, a set of phoneme classification models optimized based on duration is constructed and integrated into a Quranic phoneme classification method. The proposed algorithm achieved outstanding accuracy, ranging from 99.87% to 100% according to the Medd type. The obtained results of the proposed algorithm will contribute significantly to Qur'anic recitation recognition models.

**Keywords:** phoneme classification; phoneme duration; pattern recognition; Arabic vowels classification; recitation recognition; Tajweed recognition; speech recognition

## 1. Introduction

In recent years, the utilization of computers in the process of second language teaching and learning gained considerable attention from researchers. Systems that use a computer in teaching a second language are called Computer-Aided Language Learning (CALL) [1]. One of these systems is a system to teach Holy Qur'an [2]. In fact, the traditional and prevalent method of teaching Tajweed rules in Qur'anic teaching and learning is face-to-face based. However, computers have become smart enough to mark the student's effort based on the predetermined answers, give instant feedback, and record the student's speech

or text, where computers are trained in advance to be able to do the above-mentioned tasks [3]; thus, it is conceivable that computers can now replace teachers. Moreover, the lack of Qur'an teachers, the difficulty of going to Qur'an teaching schools in many cases, and the trend to distance learning in the last years, especially after the emergence of the COVID-19 pandemic, increase the need to develop Computer-Aided Pronunciation Learning (CAPL) systems for the Holy Qur'an.

According to [4], in his analysis study of students' recitation mistakes, the percentage of mistakes in Medd rules was 14%, which is the second-largest percentage of the mistakes in Tajweed rules; therefore, it is important to improve the level of recognition of Medd by a new model to measure the phoneme duration, especially vowels. As known, the Medd rule is one of the Tajweed rules that depend on the phoneme duration; therefore, it is necessary to measure the duration of the phoneme in order to recognize the Medd rule in Holy Qur'an recitation.

The duration can be defined as "the physical property representing the measured length of a speech sound from the perspective of the articulatory and acoustic points". On the other hand, "the speech duration may also be in the representation of the time dimension of an acoustical signal" [5].

It should be noted here that the vowels are one of the important differences between normal Arabic speech and Qur'an recitation, where the Tajweed rules include the types of Medd as the Medd expresses the amount of time the reciter (reader) takes to pronounce the vowel, and if the vowels in the normal Arabic speech are divided into a short vowel (Harakah), and a long vowel (2 Harakah) so added to that in the Qur'an recitation (4 Harakah, 5 Harakah, 6 Harakah), this is called Medd in Tajweed rules.

For given correct diacritics, the speech recognition model for Arabic usually has a simple one-to-one mapping between orthography and phonetic transcription. To accommodate short and long vowels, including the emphatic vowels [6], 14 vowels were used (/AE/, /AE:/,/AA/, /AA:/, /AH/, /AH:/, /UH/, /UW/, /UX/, /IH/, /IY/, /IX/, /AW/, /AY/).

In the Arabic language, each syllable starts with a consonant followed by a vowel that is limited and easy to detect. Short vowels are marked "V," and long vowels are marked "V:" [6]. Such syllables can be classified according to the syllable length, which is also known as Harakattes [7,8]; that syllable is represented as a consonant-vowel (CV) pattern from a discrete set specific to Arabic: CV, CV:, CVC, CV:C, CVCC, and CV:CC. This set of CV patterns is also defined as syllable weights [9].

Although the Holy Qur'an is written in Arabic, there are differences between Qur'an recitation and regular speech. Therefore, several difficulties begin when dealing with Qur'an recitation specialties because of these variations between the Qur'an written and recited [10].

The recitation of the Qur'an is characterized by the so-called Tajweed, where Tajweed is defined as a set of rules that control how the Qur'an must be recited. One of the most important Tajweed rules regarding the characteristics of Qur'anic recitation is the rule of Medd, which is lengthening the sound with a letter of the Medd. The Medd letters are: The Alif (ا) Sakinah preceded by a Fathah, The ya' (ي) Sakinah preceded by a Kasrah, and the Waw (و) Sakinah preceded by a Dhummah [11], for example, "نُوحِيهَا."

As shown in Table 1. The Medd types can be categorized in terms of their duration into three categories.

The first category is the normal prolongation of two vowels, including Medd 'asli, which is defined as the lengthening without which the letter of Medd cannot exist, and it does not follow by Hamzah or Sukoon; this type involves the Lesser Connecting Medd and The Substitute Medd. The second category is the prolongation of four vowels for Medd Wajib Mutasil and six vowels for Medd Lazim Kalimi and Medd Lazim Harfi. The third category is optional Prolongation includes Medd Jayiz Munfasil (some reciters Prolongate it with four vowels and some with two vowels) and Medd Earid Lilsukoon (with optional Prolongation two, four, or six vowels).

**Table 1.** Types of Medd.

| Type of Medd | Name of Medd | Hokom | Example |
|---|---|---|---|
| The Natural Lengthening | Medd 'asli | Normal prolongation of 2 vowels. | قَالَ |
| | Lesser Connecting Medd | Normal prolongation of 2 vowels. | إِنَّهُ كَانَ |
| | The Substitute Medd | Normal prolongation of 2 vowels. | شَكُورًا |
| The Secondary Medd | Medd Wajib Mutasil | Prolongation of 4 vowels. | السَمْاءِ |
| | Medd Jayiz Munfasil | Prolongation of 4 vowels. | أَيُّهَا اىَ |
| | The Exchange Medd | Normal prolongation of 2 vowels. | ءَادَمُ |
| | Greater Connection Medd | Prolongation of 4 vowels. | يَرَهُ أَحدٌ |
| | Medd Earid LilSukoon | Prolongation of 2 or 4 or 6 vowels. | العَالَمَيــنَ |
| | The Leen Medd | Prolongation of 2 or 4 or 6 vowels. | خَــوفٍ |
| | Medd Lazim Kalimi | Prolongation of 6 vowels. | الطَامَة |
| | Medd Lazim Harfi | Prolongation of 6 vowels. | حــم |

This classification of the due lengthening of long vowels in Tajweed, as well as vowels and consonants that are referred to as without Medd, requires the development of a new method to classify the phoneme in Qur'anic recitation. Accurate classification of vowels will help significantly in recognizing the rules of Medd, thus improving the accuracy of recognizing Qur'anic recitation.

The rules-based approach proved to be effective and low-cost when applied in improving some pattern recognition applications [12], which gives us a strong indication of the significant impact of this approach in improving the classification of phonemes in Quran recitation with a limited set of rules.

Thus, this paper aims to improve the classification of phonemes in the Holy Qur'an based on the Rules of the length of the Qur'anic recitation, which are called Medd rules. This is done by presenting a rule-based phoneme duration algorithm to improve HMM acoustic model.

The paper is structured as follows: in Section 2, a literature review of phoneme classification techniques and approaches is given. In Section 3, our proposed work is discussed in detail. In Section 4, we discuss the performance of the proposed algorithm, and our experimental results are presented, followed by a conclusion in Section 5.

## 2. Literature Review

In recent decades, researchers have tended to develop computer-aided pronunciation learning systems for the Arabic language and Qur'anic recitation. However, it is noted through a literature review that most of the works do not focus on the rules of Tajweed, which distinguishes Qur'anic recitation from normal speech, due to the fact that these works of literature attempt to adapt the techniques used in speech recognition to fit Qur'an recitation. Despite this, there are some attempts to contribute to recognizing Qur'an recitation for a variety of purposes, such as recognizing the correct pronunciation [13], correcting errors in memorizing [14], identifying some rules of Tajweed (including Tajweed rules related to letter characteristics [15] such as Izhar and Idgham, Imalah, Qalqala), and developing a Qur'anic learning system [16]. Despite the importance of Medd's rules in Qur'anic recitation, a limited number of research studies have been carried out on Qur'anic recitation and Tajweed recognition.

### 2.1. Motivation for Phonemes Classification Based on Their Duration

Speech recognition consists of converting input sound into a sequence of phonemes, then finding text for the input using language models. Therefore, phoneme classification

performance is a critical factor for the successful implementation of a speech recognition system. However, the correct distinction between phonemes with similar characteristics remains a difficult problem even for current classification methods, especially in Qur'anic recitation in which it is difficult to distinguish some similar characteristics which only specialists in the rules of Tajweed may be able to distinguish between them. For example, vowel recognition mainly depends on the classification of phonemes in the language according to the duration required for their pronunciation. Some studies have focused on classifying vowels, but it is not sufficient for the classification and recognition of phonemes in Qur'anic recitation. Where most researchers in previous studies classify the phoneme according to the classification of the English language into consonants and vowels, or voiced, unvoiced, silence, and so on, classifications that do not reflect the classification of vowels in Qur'anic recitation.

### 2.2. Arabic Phonemes Classification

Researchers' attempts to recognize and classify vowels began as early as the 1980s; for example, Pal et al. [17] described an adaptive model of a self-supervised learning algorithm with a classifier using the fuzzy set to classify English vowels. Alghamdi [18] also proved through spectrographic analysis that the phonemic implementation of the standard Arabic vowel system varies in different dialects, as the study was conducted on different Arabic dialects, including Saudi, Egyptian, and Sudanese dialects.

There have been many classification methods in the previous research, some of which categorized Arabic phonemes into two main categories: vowels and consonant phonemes, as in the research conducted by Natarajan and Jothilakshmi [19].

Greenwood and Kinghorn [20] were concerned with labeling the speech samples based on phoneme classification into silence, voiced, or unvoiced speech, using calculations over the speech samples; zero-crossing, and short-term energy functions. They obtained a classification accuracy of 65% because background noise causes the cut-off for silence to be raised, as it may not be quite zero due to noise being interpreted as speech by the functions. Also, different volumes and speeds of talking cause problems identifying voiced/unvoiced speech.

Another work by Almisreb, Abidin, and Tahir [5] discovered vowel overlapping between short vowels and their long counterpart vowels. Furthermore, the duration of Arabic vowels is investigated, with duration analysis for all vowels discussed first, followed by a study for each vowel separately. Additionally, a comparison of long and short vowels, as well as a comparison of high and low vowels, is offered.

The Arabic language does not contain words that have the same pronunciation and differ in writing or meaning. In other words, every word in the Arabic language has a unique pronunciation and writing when diacritics are taken into account. Therefore, Almekhlafi et al. [21] focused on classifying the phonemes of the Arabic alphabet with diacritics, given the importance of diacritics, and they created an Arabic dataset called Arabic alphabet phonetics dataset (AAPD) using deep neural networks and Mel-frequency Cepstral Coefficient (MFCC) for extracting features.

In order to enhance phoneme recognition, some studies have improved classification and rely not only on classification into vowels and consonants but classification based on the phonemic characteristics of the language; for example, Oh et al. [22] proposed a hierarchical phoneme clustering method to exploit more suitable recognition models to different phonemes, which is improved the performance of recognition the fricative, affricate, stop, and nasal sounds, where the Arabic phoneme was classified into 12 different classes which are Voiceless Stop, Voiced Stop, Voiceless Fricative, Voiced Fricative, Glide, Nasal, Liquid, Affricate voiced, Hamza, Fatha, Kasra, Damma.

More recently, Zangar et al. [23] conducted a study of the period model in the Arabic language by developing a variety of DNN-based structures, taking into account some characteristics of Arabic, i.e., vowel quantity (short vs. long vowels) and gemination; thus,

phonemes were classified to short vowels, long vowels, simple consonants, geminated consonants in addition to pauses.

In addition, some researchers used other approaches to classify the phoneme based on time. For example, Ircio et al. [24] proposed a filter method to select a subset of time series for the purpose of supervised classification of multivariate time series.

In order to discover pronunciation errors in Arabic words, Nazir et al. [25] used two different methods, CNN Features-based and k-nearest neighbor (KNN) with support vector machine (SVM), to classify Arabic phonemes into 28 phonemes. For the same purpose, [26] has improved error detection by developing an algorithm to select the most distinct set of features to improve the detection of pronunciation errors using CNN.

Recently, Asif et al. [27] proposed a classification of Arabic phonemes to increase the accuracy of identifying Arabic phonemes with diacritics (short vowels). Since the Arabic consonants have three possible states, which are consonants with the three short vowels, and they make a total of 84 unique phonemes, so they presented a classification of Arabic phonemes using the deep convolutional neural network (CNN) into 84 classes.

Figure 1 shows the chronology of the research that has been done regarding the classification of phonemes in the Arabic language and Qur'anic recitations. The figure also clarifies the types of phoneme classifications presented by previous researchers, where some researchers classified the Arabic phoneme into consonant and vowel, others increased the distinction between long vowels and short vowels, and other research divided phonemes into phonemic, non-vocal, and silent. At the same time, others classified the phoneme according to the characteristics of the letters or by focusing on the diacritics that characterize the Arabic letter.

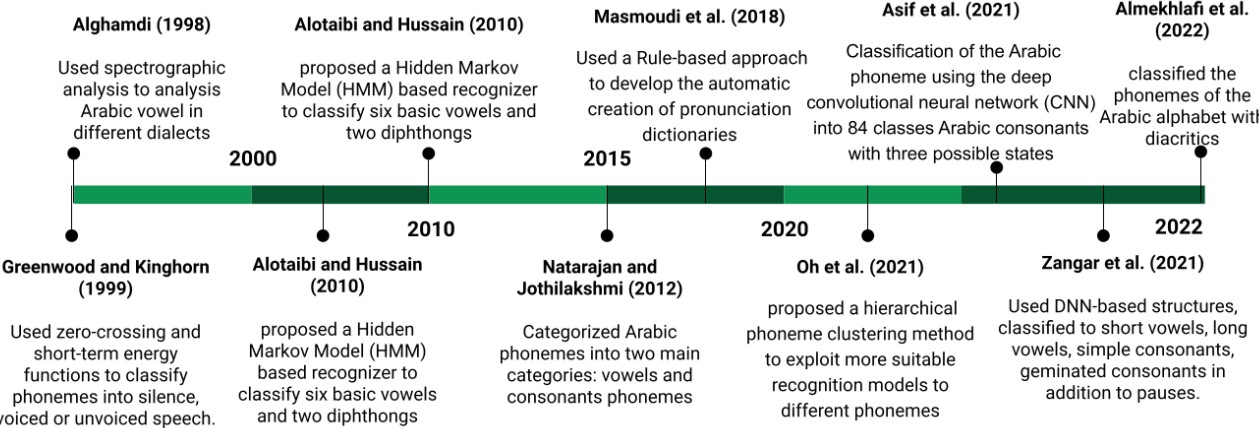

**Figure 1.** Chronology of the research related to Arabic phonemes classification.

### 2.3. Rule-Based Classification Approach

Although the majority of the opinions quoted in the literature claims that the rule-based technique is inefficient since it requires a large number of rules for the detection of phoneme boundaries [28], there are some researchers suggesting rule-based methods yield improved phoneme segmentation with a small number of rules, demonstrating that it is cost-effective.

Ramteke and Koolagudi [12] proposed a rule-based method for segmenting and classifying speech signals into phonemes automatically. They employed the Pitch and zero-frequency filter signal to determine the region of change from voiced to unvoiced and vice versa and then proposed a finite set of rules based on the power spectrum differences seen during phoneme transitions. This prompts researchers to use a rule-based approach to improve phoneme classification.

Masmoudi et al. [29] also used a rule-based approach to develop the automatic creation of pronunciation dictionaries due to the relationship between orthography and pronunciation being relatively regular and well-understood in the MSA language. This research

confirms that although a rule-based approach requires specific linguistic skills to design the rules, the rule-based approach produces good-quality pronunciation dictionaries without requiring a great amount of data.

It is worth mentioning that previous research has taken different approaches to classifying the Arabic phoneme. For example, Newman and Verhoeven [30] perform an examination study of formant frequencies of Arabic vowels. This work was focused on Qur'an recitation tokens, consisting of a Qur'an recitation of 30 min. The token recorded contains 400 vocal observations, covering all the vowels in Arabic.

Iqbal et al. [31] presented a preliminary study aimed at segmenting and identifying Arabic vowels using formant transitions in the continuous recitation of the Qur'an. Their algorithm recognized the vowels based on the extracted and pre-segmented audio file formats of the recitation and achieved an accuracy rate of up to 90% in the Qur'anic recitation files, which include about 1000 vowels.

Alotaibi and Hussain [32] proposed a Hidden Markov Model (HMM) based recognizer to classify vowels (six basic vowels and two diphthongs). These vowels have been analyzed in both time and frequency domains using a spectrogram technique to help understand the similarities and dissimilarities between the phonetic features of vowels.

Tolba et al. [33] also proposed an algorithm for the segmentation of Arabic consonant and vowel speech without linguistic information using wavelet transformation and spectral analysis and focused on the transient search between consonants and vowels at certain levels of wavelet decomposition, and this study achieved an accuracy rate of 88.3% for Consonant/Vowel Segmentation.

Natarajan and Jothilakshmi [19] classified speech into vowels and consonants using Formants and support vector machines (SVMs) for the purpose of segmenting speech into smaller units.

Sarma and Sarma [34] used a combination of three types of ANN structures, namely recurrent neural network (RNN), SOM, and probabilistic neural network (PNN), to recognize and classify the phonemes. RNN is used to classify the incoming words into six unique Assamese phoneme families based on the initial phoneme, and a SOM and PNN-based phoneme segmentation and recognition algorithm are used to segment and recognize the initial phoneme.

Zangar, Mnasri, Colotte, and Jouvet [23] proposed DNN models to be trained on different sets of phonemes, namely the DNN-all-phone model that was trained on all the phonemes, whereas class-specific DNN models were trained separately on each phoneme class.

Modern Standard Arabic (MSA) has 34 phonemes: three short vowels /a, i, u/, three long vowels /a:, i:, u:/, and 28 consonants, which in turn increase the complexity Algorithmic recognition algorithm. Thus, Ibrahim et al. [35] have integrated Genetic Algorithms (GAs) into various speech-processing applications to reduce the computational complexity of the recognition algorithm while improving the accuracy of distinctive phonetic features (DPF) recognition. Oh, Park, Kim, and Jang [22] also attempted to reduce the complexity by using a hierarchical approach to classify Arabic phonemes into 12 categories.

Despite the diversity of these approaches, the rules-based approach can be of great use in improving the classification of Arabic phonemes in the Holy Qur'an, especially with regard to long vowels.

### 2.4. Trends and Research Directions

In general, most of the literature related to the Arabic language attempts to classify phonemes similarly to other languages' classification, classifying phonemes into consonants and vowels or into voiced, unvoiced, and silence. Some literature also attempted to classify phonemes based on the Arabic language characteristics. For example, the study conducted by Zangar, Mnasri, Colotte, and Jouvet [23] proposed to classify Arabic phonemes into four categories based on the type of phoneme and its characteristics in the Arabic language. Some studies also focused on short vowels and their impact on speech recognition, such as

studies performed by Almekhlafi, Moeen, Zhang, Wang, and Peng [21,22,27], and some discussed a distinction between short and long vowels [35].

Despite these efforts and attempts, the previous literature ignored the rules of Medd, which constitute an additional classification in the Qur'anic recitation, that makes the classifications of Arabic phonemes in the previous studies inadequate and not applicable to the insufficient to study the rules of Tajweed, as it ignores phoneme features of recitation. For instance, the phoneme Medd has detailed rules in Tajweed recitation that must be taken into account. However, the Medd has limited features in Arabic language as the Medd is classified in the Arabic language in to long vowels and short vowelswhereas in Tajweed recitation, the Medd phoneme has more features that are totally beyond Arabic language phoneme characteristics. Through this short review, we can conclude that the research focusing on duration modeling in Qur'anic recitation and the phonemes classification based on it still needs further study.

Therefore, this paper aims to cover this gap attempting at the same time to improve the HMM model with a rule-based approach to get a more accurate classification of Qur'anic long vowels based on phoneme duration in Tajweed rules.

### 3. Procedures and Methods

To improve Qur'anic phoneme classification by using a rule-based phoneme duration algorithm, we used Hidden Markov Toolkits (HTK) for implementation. There are three basic stages in this work, which are data collection and dataset construction, building the acoustic model using HMM model, and phoneme classification in Qur'anic recitation based on the rules of Tajweed. These basic stages and their procedures are illustrated in Figure 2.

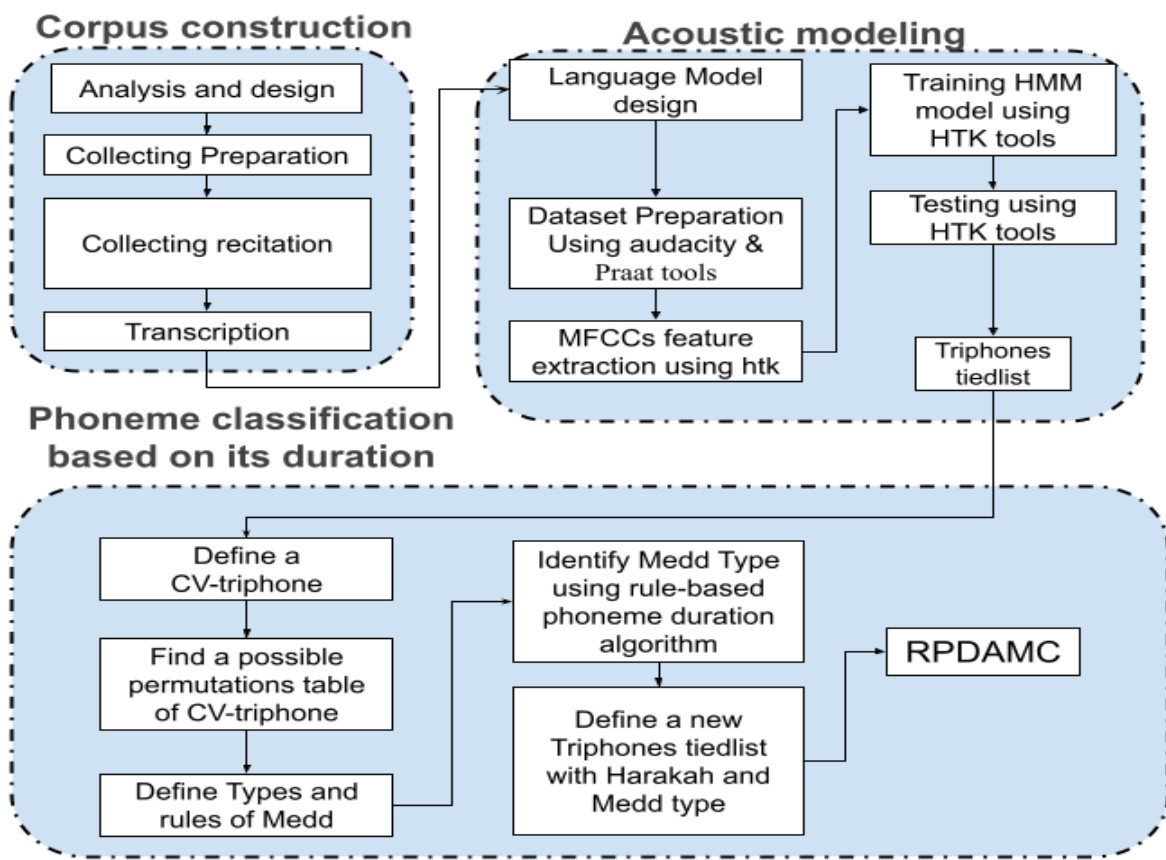

**Figure 2.** Stages of classification of phonemes based on the rules of Tajweed.

### 3.1. Corpus Construction (Collect and Design Dataset)

Collecting and building a sufficient and suitable database is one of the most basic steps in the field of speech recognition and its applications. This sub-section discusses the steps of building the database used in the training and testing phase of this work.

#### 3.1.1. Analysis and Design

Data preparation is the first stage of any recognizer development project because spoken data are required for both training and testing. Thus, optimum speech corpus selection is a significant factor affecting the recognizer's efficiency [36]. This entails finding samples of speech data that have parametrical properties representative of the speech variability required for the application task. Therefore, the choice of the reciters has been done with care and attention, and the diversity of the gender, nationality, and age of reciters, as well as skills of recitation, was taken into consideration. Nevertheless, the data collected was also reviewed by an expert specialized in Qur'an recitation to ensure that the recitation conforms to the Tajweed rules.

Therefore, the dataset had to be designed, and we chose 21 Qur'anic verses, shown in Table 2 that include all types of Medd (long vowels) in the Qur'an as well as all letters of Medd for each type. In addition, these verses include all the Arabic phonemes. At this stage, criteria were also prepared for selecting the reciters whose recitations we will collect.

**Table 2.** The verses selected to the dataset.

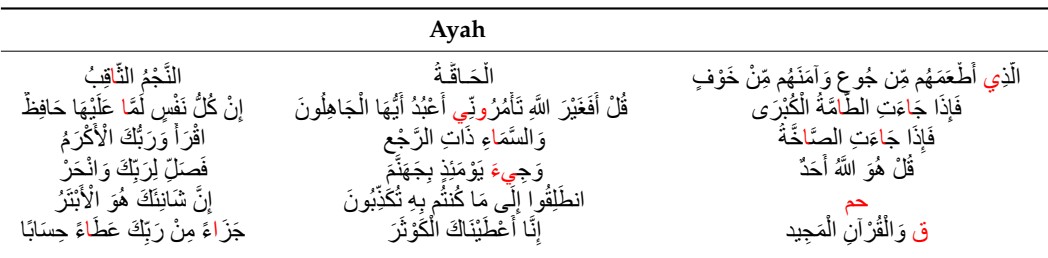

| | Ayah | |
|---|---|---|
| النَّجْمُ الثَّاقِبُ | الْحَاقَّةُ | الَّذِي أَطْعَمَهُم مِّن جُوعٍ وَآمَنَهُم مِّنْ خَوْفٍ |
| إِنْ كُلُّ نَفْسٍ لَمَّا عَلَيْهَا حَافِظٌ | قُلْ أَفَغَيْرَ اللَّهِ تَأْمُرُونِّي أَعْبُدُ أَيُّهَا الْجَاهِلُونَ | فَإِذَا جَاءَتِ الطَّامَّةُ الْكُبْرَى |
| اقْرَأْ وَرَبُّكَ الْأَكْرَمُ | وَالسَّمَاءِ ذَاتِ الرَّجْعِ | فَإِذَا جَاءَتِ الصَّاخَّةُ |
| فَصَلِّ لِرَبِّكَ وَانْحَرْ | وَجِيءَ يَوْمَئِذٍ بِجَهَنَّمَ | قُلْ هُوَ اللَّهُ أَحَدٌ |
| إِنَّ شَانِئَكَ هُوَ الْأَبْتَرُ | انطَلِقُوا إِلَى مَا كُنتُم بِهِ تُكَذِّبُونَ | حم |
| جَزَاءً مِنْ رَبِّكَ عَطَاءً حِسَابًا | إِنَّا أَعْطَيْنَاكَ الْكَوْثَرَ | ق وَالْقُرْآنِ الْمَجِيدِ |

#### 3.1.2. Collecting Preparation

After analysis and design, it was essential to first collect speech data from Hufaz (reciters memorizing the Qur'an), segment Ayat and label each audio file. Thirty reciters (speakers) were asked to record their recitations of 21 selected Ayahs in a suitable location. Each reciter was instructed to recite each Ayat in four different styles: normal, quick, slow, and incorrect recitation.

#### 3.1.3. Collecting Recitation and Preprocessing

The recitation of 30 reciters were recorded in different environments. Thus, the recorded data need to be further edited to remove unwanted segments such as interlaced sounds, noise, repeat sentences, long pauses, etc. So, the manual segmentation method was used since it is more exact, more accurate, and controlled, making it better suited for developing a recognizer's training dictionary. The selected verses were segmented, noise, and long pausing, and duplicates were removed using Audacity 2.3.2, as shown in Figure 3.

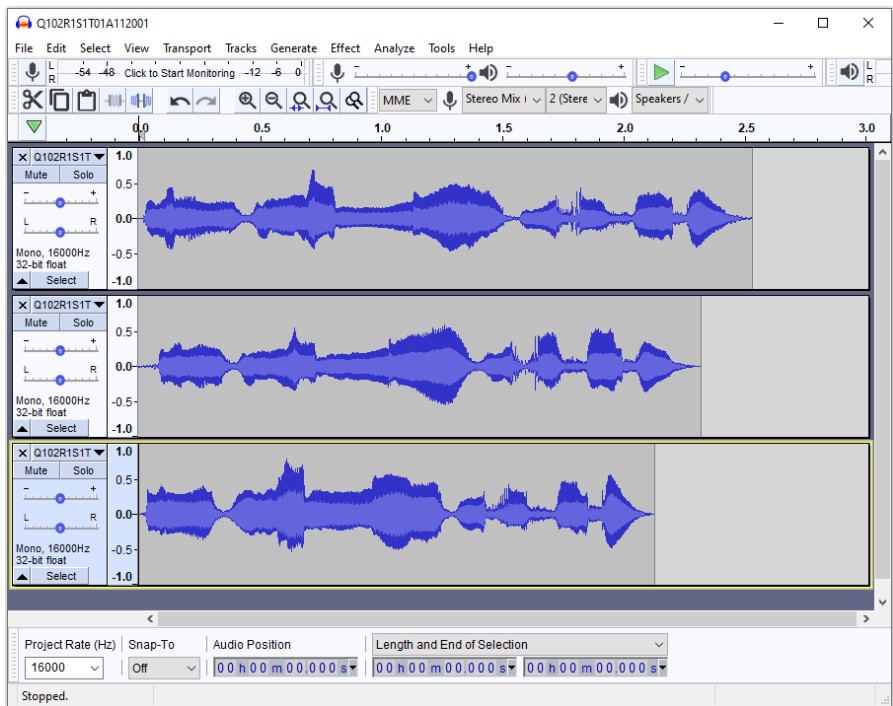

**Figure 3.** Using the Audacity program for recording and pre-processing.

The same software was used for the sampling rate, and there are many audio files recorded at a high sampling rate: 16 kHz, 22 kHz, and 48 kHz [37]. Some audio files have a lower sampling rate, 8 kHz. Because of this difference, it was necessary to standardize the rate; then, audacity software was used to standardize the rate at 16 kHz.

### 3.1.4. Transcription

Creating a sorted list of the words contained in the grammar (one per line) with pronunciations is usually the first step in developing the Pronunciation Dictionary. Arabic phonetic dictionaries [38] were used in order to permit HTK to compile an Acoustic Model. Figure 4 shows a sample of Arabic phonetic dictionaries for this work dataset.

| | change.log | wdnet | triphones1 | prompts | monophones0 | dict | fulllist0 | phones.mlf | dlog | triphones1 | wintri.mlf |
|---|---|---|---|---|---|---|---|---|---|---|---|
| 1 | [ اقْرَأ ] | | | اقْرَأ | | | e q r aa e sp | | | | |
| 2 | [ الْأَبْتَرُ ] | | | الْأَبْتَرُ | | | e l e ae b t ae r ux sp | | | | |
| 3 | [ الْأَكْرَمُ ] | | | الْأَكْرَمُ | | | e l e ae k r aa m uh sp | | | | |
| 4 | [ الْجَاهِلُونَ ] | | | الْجَاهِلُونَ | | | e l jh ae: h ih l uw n ae sp | | | | |

**Figure 4.** Sample of Arabic Dictionary.

As this dictionary generated a basis for the Arabic language, we had to add some Qur'an words, which are the words that represent the Medd Lazim Harfi. This is because these letters are not pronounced as written but are pronounced with three letters, of which the second letter is a long vowel. In the dataset in this work, three letters represent the Medd Lazim Harfi which are: حم, ق, and ن.

Figure 5 shows an error in expressing these letters in the Arabic phonetic dictionaries; as it was considered as the regular letters of ح م, the two letters were considered the letter H and the letter M, while it was pronounced in the Qur'an recitation as حا ميم, which means that it is pronounced with five Arabic letters, therefore ([حم] hh ae: m iy m sp ) was added to the dictionary. Likewise, the word ق was considered in the dictionary as one letter, but it is pronounced in the Qur'an recitation قاف, which is three letters, therefore ([ق] q aa:

f sp) was added to the dictionary. Also, the word نٍ is pronounced by the Qur'an recitation, نون which is three letters, and therefore ([نٍ] n uw n sp) was added to the dictionary.

| 76 [نْ] | نْ | n w n sp |
|---|---|---|
| 47 [حمْ] | حمْ | hh m sp |
| 65 [قْ] | قْ | q f sp |

**Figure 5.** Medd Lazim Harfi in Arabic phonetic dictionaries.

The next step aims to transcribe the prompts to word Level Transcriptions and Phone Level Transcriptions because HTK cannot process the prompts file directly. Therefore, word Level Transcriptions were done using the Master Label File (MLF) approach.

Figure 6 shows the format of the MLF file on the left side.

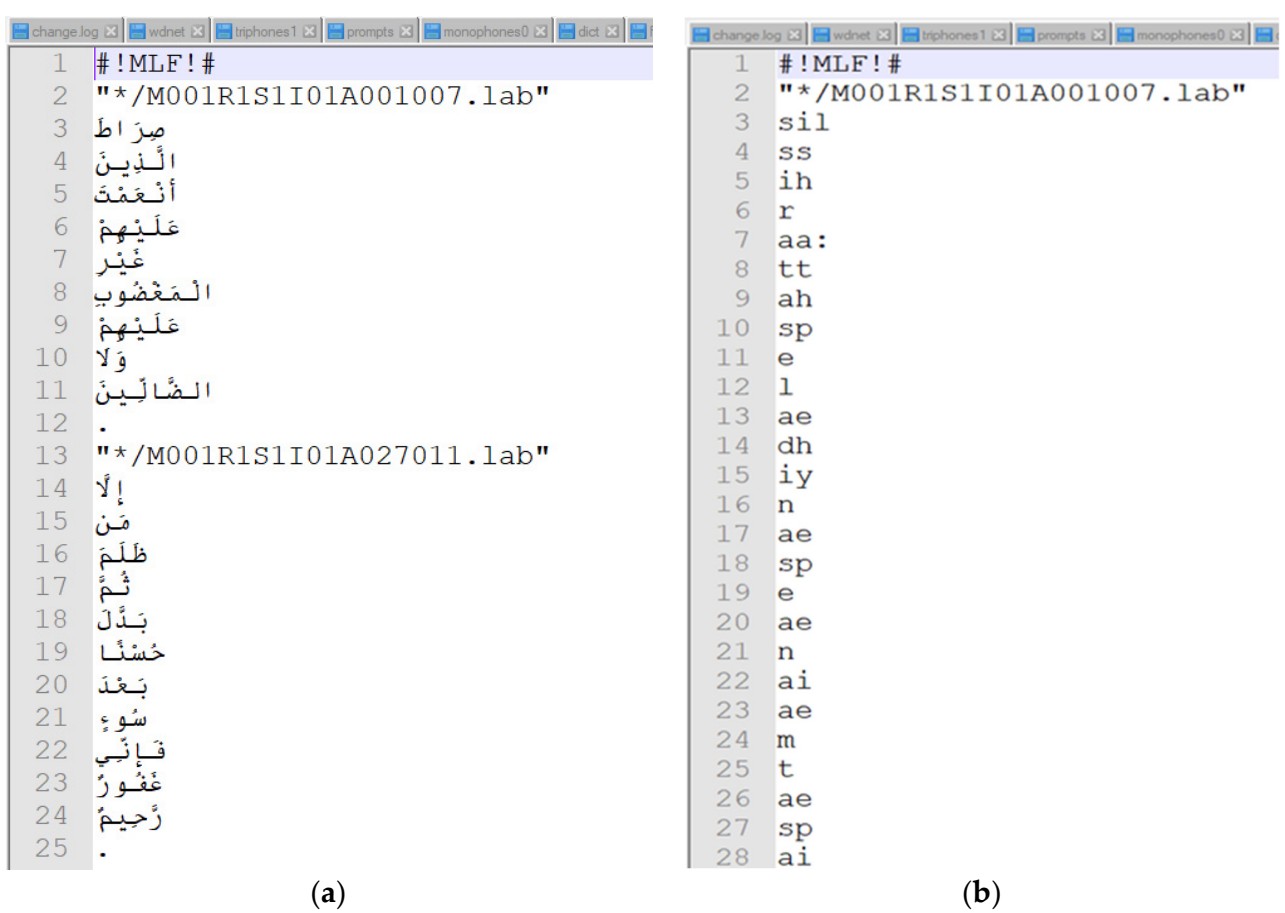

(**a**)　　　　　　　　　　　　　　　　　　　　　　　(**b**)

**Figure 6.** Sample of Level Transcriptions, (**a**) words Level Transcriptions, and (**b**) Phone Level Transcriptions.

Then, word Level Transcriptions were expanded to Phone Level Transcriptions by replacing each word with its phonemes, thus producing a new Phone Level Master Label File, as shown in Figure 6 on the right side.

### 3.2. Acoustic HMM Model Using HTK Toolkit

To classify phonemes based on their duration, HTK has been used to generate HMM triphones, where a new rule-based algorithm was proposed to improve HTK Tied-State triphones. There are 4 sub-steps that will be detailed in this section.

### 3.2.1. Language Model Design and Dataset Preparation Using Audacity and Praat Tool

The grammar or Language Model is one of the main components of any Speech Recognition model. The model contains a large list of words and their probability of occurrence in a given sequence. In the case of Qur'an recitation recognition models, the sequence of words is fixed because the sequence must be adhered to in the Holy Qur'an, but stops and repetition should be considered.

In this paper, the dataset of Qur'an recitation recognition models, Audacity 2.3.2 and Praat applications are used to prepare the audio datasets, shown in Figures 3 and 7. Praat is an open software tool used by many audio researchers for various acoustic analyses. It is free and available on many platforms. Praat supports audio files with .wav extensions [39].

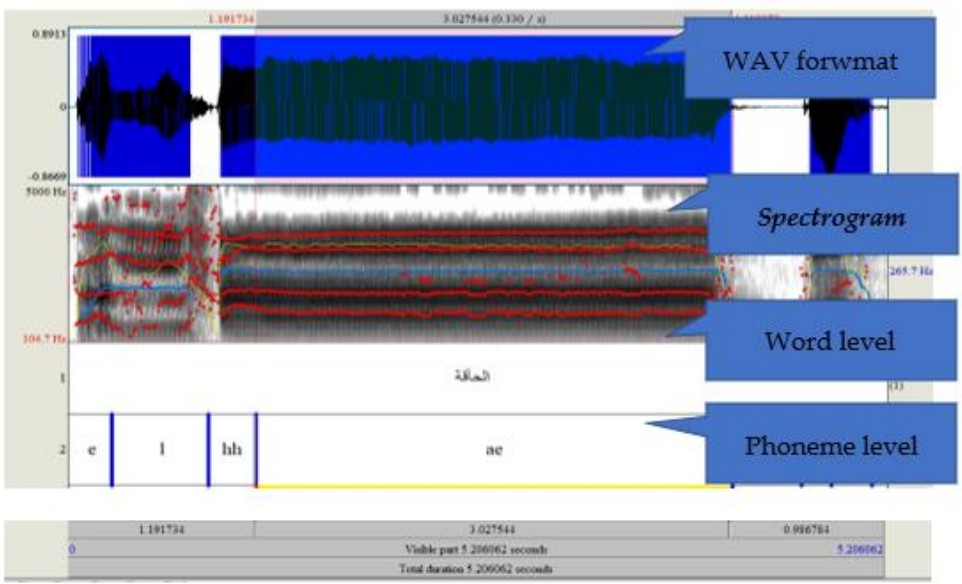

**Figure 7.** Praat Editor window.

### 3.2.2. Features Extraction

The goal of feature extraction is to find a set of properties of an utterance that have acoustic correlations in the speech signal, i.e., parameters that can somewhat be estimated through the processing of the signal waveform. Such parameters are termed as features [40].

Mel-Frequency Cepstral Coefficients is the most popular and evident feature extraction technique. Used extensively in speech recognition systems, Mel Frequency Cepstral Coefficients (MFCC) was used to convert audio .wav files to another format called MFCC format, which is referred to as "feature vectors"; moreover, it approximates the response of the human system more closely than any other system because logarithmically, frequency bands are positioned here. MFCC is based on signal decomposition with the help of a filter bank, which uses the Mel scale [41]. This is done in steps and processes, as shown in Figure 8:

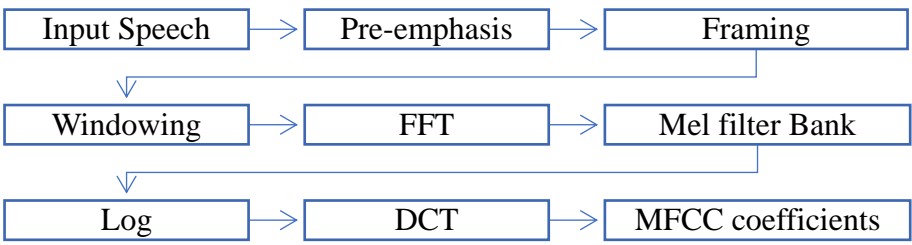

**Figure 8.** MFCC Feature extraction.

First, the speech signal is divided into time frames which consist of an arbitrary number of samples. In most systems, frame overlap is used to smooth frame-to-frame transitions. With Hamming window, each time frame is then windowed to remove discontinuities at the edges. Then to speed up the processing, Fast Fourier Transformation (FFT) is calculated for each frame to extract frequency components of a signal in the time domain. The logarithmic Mel-Scaled filter bank is applied to FFT, and then the MFCC results on Discrete Cosine Transform (DCT) of a real logarithm of short-time energy expressed in the Mel frequency scale. Finally, MFCC can be computed by using the Formula (1) [42,43]:

$$Mel\ (f) = 2595 * \log10\ (1 + f/700). \tag{1}$$

These configuration parameters within HTK Hcopy.conf followed in the description of the MFCC extraction process, pre-emphasizing the signal, frame blocking and Hamming Windowing, Filterbank, and MFCC coefficients. Figure 9 demonstrates the setting of those configuration parameters relating to the extraction process of such MFCC [44].

| Configuration parameters | Value | MFCC extraction process |
|---|---|---|
| SOURCEKIND | waveform | |
| SOURCRATE | 625 | |
| TARGETKIND | mfcc_0 | |
| PREEMCOEF | 0.97 | Pre-emphasis |
| TARGETRATE | 100,000 | Frame blocking and Hamming windowing |
| WINDOWSIZE | 250,000.0 | |
| USEHAMMING | TRUE | |
| NUMCHANS | 24 | Filterbank and MFCC coefficients |
| NUMCEPS | 12 | |

**Figure 9.** Configuration file parameters.

### 3.2.3. Training and Testing the HMM Model

The next step for feature extraction is to build a prototype description to describe the topology required for each HMM. HTK lets HMMs construct with any desired topology. A multivariate Gaussian distribution was used to train the acoustic models, where each variable is represented by one of the MFCC features. The Baum-Welch algorithm was used by HTK to estimate and optimize the model's means and covariances. This algorithm is similar to the EM algorithm. Figure 10 shows the processes of training using HTK.

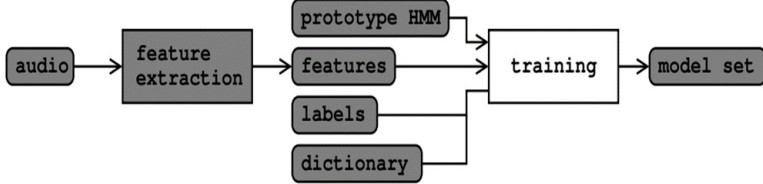

**Figure 10.** Block Diagram of the Training Processes.

### 3.2.4. Creating Tied-State Triphones

The final stage of acoustic model building is to create context-dependent triphone HMMs. This stage includes two steps: Making Triphones from Monophones and Making Tied-State Triphones. Therefore, this task begins with the conversion of Monophone transcriptions to triphone transcriptions, followed by the creation of a collection of Triphone models by copying and re-estimating the monophones. Then, similar acoustic states of

these triphones are linked to ensure that all state distributions are accurately estimated. Figure 11 shows a sample of the resulting triphone tree.

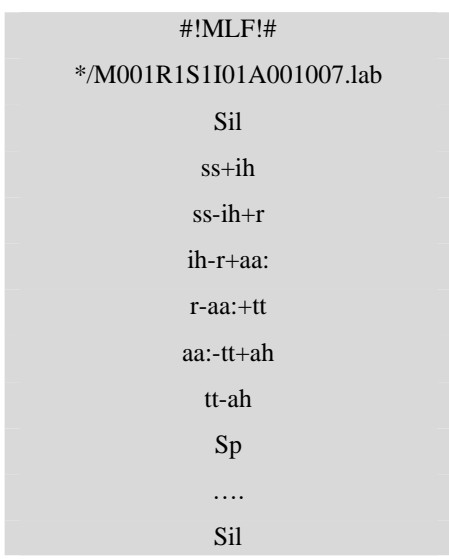

#!MLF!#

*/M001R1S1I01A001007.lab

Sil

ss+ih

ss-ih+r

ih-r+aa:

r-aa:+tt

aa:-tt+ah

tt-ah

Sp

….

Sil

**Figure 11.** Sample of HTK triphone tree.

Finally, WER was measured by the HTK tool called "HResults" in order to evaluate the recognition model using the training data collected. HResults compares the file recout.mlf, which contains the recognition results, and the file testref.mlf, which is the corresponding reference copies.

Figure 12 shows the result of the recognition accuracy assessment for the line starting with SENT, showing the number of sentences recognized where there were 252 test sentences, and 83.73% were correctly recognized. For the line beginning with WORD showing word recognition, there were 1602 words in total, of which 98% were correctly recognized. However, because the model recognized words that were not in the audio file (i.e., insertion errors), it only gets a 93.32% accuracy rating. In addition, "D" indicates Deletion Error, "S" indicates Substitution Error, and "I" indicates Insertion Error.

```
C:\Segmentation\test>HResults -I testref.mlf tiedlist recout.mlf
===================== HTK Results Analysis =====================
  Date: Sat Jul 24 00:06:46 2021
  Ref : testref.mlf
  Rec : recout.mlf
------------------------ Overall Results ------------------------
SENT: %Correct=83.73 [H=211, S=41, N=252]
WORD: %Corr=98.00, Acc=93.32 [H=1570, D=3, S=29, I=75, N=1602]
================================================================
```

**Figure 12.** HTK results or Qur'anic Tajweed acoustic model.

*3.3. Phoneme Classification Based on Its Duration Phase*

Phoneme classification based on its duration is a crucial step in recognizing the Quranic recitation, as many of the rules governing recitation depend on the duration of the letter. The process of improving the phoneme classification generated by the HMM-based model goes through steps starting from CV-triphone analysis and finding possible cases, then identifying the type of Medd based on the rules of Tajweed in the Qur'anic recitation, and finally applying our rules-based algorithm to classify the phoneme according to the duration.

### 3.3.1. CV-Triphone and Possible Permutations Table

Triphones are context phonemes (basis phoneme P with the left and right context: Pleft − P + Pright), which can be obtained from the acoustic model that was designed in the previous stage using HTK.

As is known, Medd rules relate to the long vowels (V:) and affect the classification of the Medd type; the phoneme type of letter preceding it, the letter following it, and in some cases, the second letter after it. This means that the Medd classification is affected by the type of phoneme, which is a long vowel and a vowel Short and a consonant, and because the letter hamza has an effect on the type of Medd, it must be considered as a special case of consonants. Therefore, it is important to classify the phonemes in the triphone tree to the phoneme type in order to reduce the number of permutations accepted in the Qur'anic recitation to facilitate its study, analysis, and definition of Medd cases. Thus all the phonemes in the triple phoneme tree have been classified based on phoneme type (CV-triphone) into six classes: long vowels (V:), short vowels (V), consonant (C), hamza (e), space between the two words (sp), and silence (Sil). Then, a possible permutations table of cases of Medd was prepared based on the Arabic grammar and Tajweed rules.

A possible permutations table of cases of Medd has been prepared to apply in the RPDAMC for defining the type of Medd through the triphone tree.

As is known, Medd rules relate to the long vowels (V:) and affect the classification of the Medd type; the phoneme type of letter preceding it, the letter following it, and in some cases, the second letter after it. This means that the Medd classification is affected by the type of phoneme, which is a long vowel and a vowel Short and a consonant, and because the letter hamza has an effect on the type of Medd, it must be considered as a special case of consonants. Therefore, it is important to classify the phonemes in the triphone tree to the phoneme type. Thus, all the phonemes in the triple phoneme tree have been classified based on phoneme type (CV-triphone) into six classes: long vowels (V:), short vowels (V), consonant (C), hamza (e), space between the two words (sp), and silence (sil).

To find out the number of possible CV-triphone tree cases, the following must be taken into consideration:

- The maximum number of words concerned to extract the Medd rule for any Medd letter is two, and there are at most four phonemes concerned by the Medd rules that are either preceding or following the letter itself. Thus, it is necessary to take into account all the permutations for LPR, Next L where "P" means the phoneme treated, "L" represents the phoneme preceding the phoneme treated, "R" represents the phoneme immediately following the phoneme treated, "Next L" represents the first phoneme in next word following the phoneme treated;
- The beginning of the word consists of P and R parts only, and the end of the word also consists of L and P parts only;
- Hamza phoneme (e) influences the classification of the Medd if it comes directly after the Medd letter or at the beginning of the next word; therefore, it is separated from the rest of the consonants;
- The letters that have Shaddah were taken into consideration because of their effect on the Medd rules;
- Since the rules of Medd are either in one word or in two words, and the Medd letter is in one word, either it is within the word in the middle of the recitation or at the end of the last word before stopping reciting (silence). Thus, it can be divided into three categories: Medd cases within one word, Medd cases at the end of the last word, and Medd cases within two words.

Given these considerations, we can get that: possible cases of L = (sp, V, V:, C), possible cases of P = (V, V:, C), possible cases of R = (sp, V, V:, C, e, sil), and Next L = (e, C, sil, *) (where V: is long vowel, V short vowel, C is consonant but not Hamza, e Hamza letter, sil is

silence, sp space between two phonemes and * means any phoneme). Thus, we can obtain the permutations formulation to find out the number of possible cases;

Permutations = (number of cases L * number of cases P * number of cases R * number of cases nex

Thus, we can find the number of possible cases for each of the three categories:

- First classification: the number of possible states of the triple phoneme within one word. In this category, the cases of L can be (V, V:, C, e, or blank), possible cases of P are (V, V:, C, or e), possible cases of R are (V, V:, C, or e), and Next L can be only (*), hence the number of possible Permutations = 5 × 4 × 4 × 1 = 80;
- Second classification: the number of possible states of the triple phoneme at the end of the last word. In this category, the cases of L can be (V, V:, C, or e), possible cases of P are (V, V:, C, or e), possible cases of R are (V, V:, C, e, or blank), and Next L can be only (sil). Hence the number of possible Permutations = 4 × 4 × 5 × 1 = 80;
- Third classification: the number of possible cases of the triple phoneme within two words. In this category, the cases of L can be (V, V:, C, or e), possible cases of P are (V, V:, C, or e), possible cases of R can be only (blank), and Next L can be only (C or e). Hence the number of possible Permutations = 4 × 4 × 1 × 2 = 32. In addition, sil and sp are special cases in the triphone tree.

So the total of the possible cases of CV-triphone tree = 80 + 80 + 32 + 2 = 164.

These cases are theoretically possible, but since the syllables in the Arabic language are confined to six patterns, which are CV, CV:, CVC, CV:C, CVCC, and CV:CC, all the unacceptable cases in the Arabic language have been canceled (which are 142 cases, for example, CCC) and 52 cases accepted for study in terms of the Medd rules. All 52 cases that have been studied and classified in terms of Medd type and its rule are shown in Table 3.

**Table 3.** Possible CV-triphone tree cases table based on Medd Rules.

| L | P | R | Next L | Medd Type | Rule | L | P | R | Next L | Medd Type | Rule |
|---|---|---|---|---|---|---|---|---|---|---|---|
| C | V: | C | * | Medd 'asli | 2 | blank | V: | V | * | No Med | 1 |
| V: | C | V | * | Medd 'asli | 2 | blank | V: | V: | * | No Med | 1 |
| e | V: | C | * | Medd 'asli | 2 | blank | e | C | * | No Med | 1 |
| C | V: | C | Sil | Medd 'asli | 2 | blank | e | e | * | No Med | 1 |
| e | V: | C | Sil | Medd 'asli | 2 | blank | V | V | * | No Med | 1 |
| C | V: | blank | Sil | Medd 'asli | 2 | blank | V | V: | * | No Med | 1 |
| e | V: | blank | Sil | Medd 'asli | 2 | C | C | C | * | No Med | 1 |
| C | V: | blank | C | Medd 'asli | 2 | C | V | e | * | No Med | 1 |
| V: | C | blank | e | Medd 'asli | 2 | V | V | C | * | No Med | 1 |
| V: | C | blank | C | Medd 'asli | 2 | e | V | C | * | No Med | 1 |
| e | V: | blank | C | Medd 'asli | 2 | e | V | e | * | No Med | 1 |
| V: | C | V | Sil | Medd Eaeid LilSkoon | 2,4,6 | C | C | C | Sil | No Med | 1 |
| C | V: | blank | e | Med Jayaz | 2,4 | C | V | e | Sil | No Med | 1 |
| e | V: | blank | e | Med Jayaz | 2,4 | V | V | C | Sil | No Med | 1 |
| V: | C | blank | Sil | Med Lazim harfi | 6 | e | V | C | Sil | No Med | 1 |
| V: | C | C | * | Med Lazim Kalmi | 6 | e | V | e | Sil | No Med | 1 |

**Table 3.** *Cont.*

| L | P | R | Next L | Medd Type | Rule | L | P | R | Next L | Medd Type | Rule |
|---|---|---|---|---|---|---|---|---|---|---|---|
| V: | C | C | Sil | Med Lazim Kalmi | 6 | C | V | blank | Sil | No Med | 1 |
| C | V: | e | * | Med Wajib | 4 | e | V | blank | Sil | No Med | 1 |
| V: | e | V | * | Med Wajib | 4 | C | C | blank | e | No Med | 1 |
| e | V: | e | * | Med Wajib | 4 | C | C | blank | C | No Med | 1 |
| C | V: | e | Sil | Med Wajib | 4 | V | e | blank | e | No Med | 1 |
| V: | e | V | Sil | Med Wajib | 4 | V | e | blank | C | No Med | 1 |
| e | V: | e | Sil | Med Wajib | 4 | V | V | blank | e | No Med | 1 |
| V: | e | blank | Sil | Med Wajib | 4 | V | V | blank | C | No Med | 1 |
| V: | e | blank | e | Med Wajib | 4 | e | e | blank | e | No Med | 1 |
| V: | e | blank | C | Med Wajib | 4 | e | e | blank | C | No Med | 1 |

Through the analysis and study of these cases, a scientific formulation has been concluded to identify the type of Medd. The details of each Medd will be discussed separately in the following subsections.

### 3.3.2. Identify Medd Type

In this research, a rule-based algorithm was used to classify the phonemes into Medd types. A rule-based model is composed of four components: a knowledge base, a rule base, an inference engine, and an execution engine [45]. The facts and conditions are described in the knowledge base. A relationship between the premises and the conclusions is described by the rule basis. The inference engine has a pattern matcher for all relevant and applicable rules. The execution engine determines which rules to apply, given the input.

Two main algorithms are used to infer a rule-based system, forward chaining, and backward chaining. This research used a forward chaining algorithm, which is a top-down approach. It is also known as a data-driven technique because it reaches the goal using the available data; thus, it starts with the facts and uses the rules to deduce the conclusions or trigger an action given the facts [46].

### 3.3.3. Rule-Based Phoneme Duration Algorithm for Medd Classification

As defined in the Tajweed rules, the duration of letters is measured by Harakah, which is the time of pronouncing a short vowel. Likewise, the Medd letter is also measured by Harakah; for example, Medd 'asli should be 2 Harakah, Medd Wajib Mutasil duration is 5 Harakah, Medd Jayiz Munfasil duration is 4 Harakah, Medd Lazim Kalimi duration is 6 Harakah, Medd Lazim Harfi duration is 6 Harakah, Medd Earid LilSukoon duration is 2, 4 or 6 Harakah [4,47].

Given these rules, a rule-based phoneme duration algorithm for Medd classification (RPDAMC) based on Medd rules has been proposed. The basic idea is to add the number of Harakah to the phoneme tree generated in the HTK triphone based on its duration in the Tajweed rules in order to use it at the recognition stage. In addition, RPDAMC classifies phonemes according to the main types of Medd. The triphone tree L-P + R, after Medd classification using the RPDAMC, will be as $L(1) - P(6) + R(1)$, where the duration of the phoneme is between brackets.

Figure 13 shows the use of the RPDAMC Algorithm to identify the Medd type. It shows all the states of the Medd and adds the rule of Medd to the triphone and how many Harakah the phoneme must be lengthened.

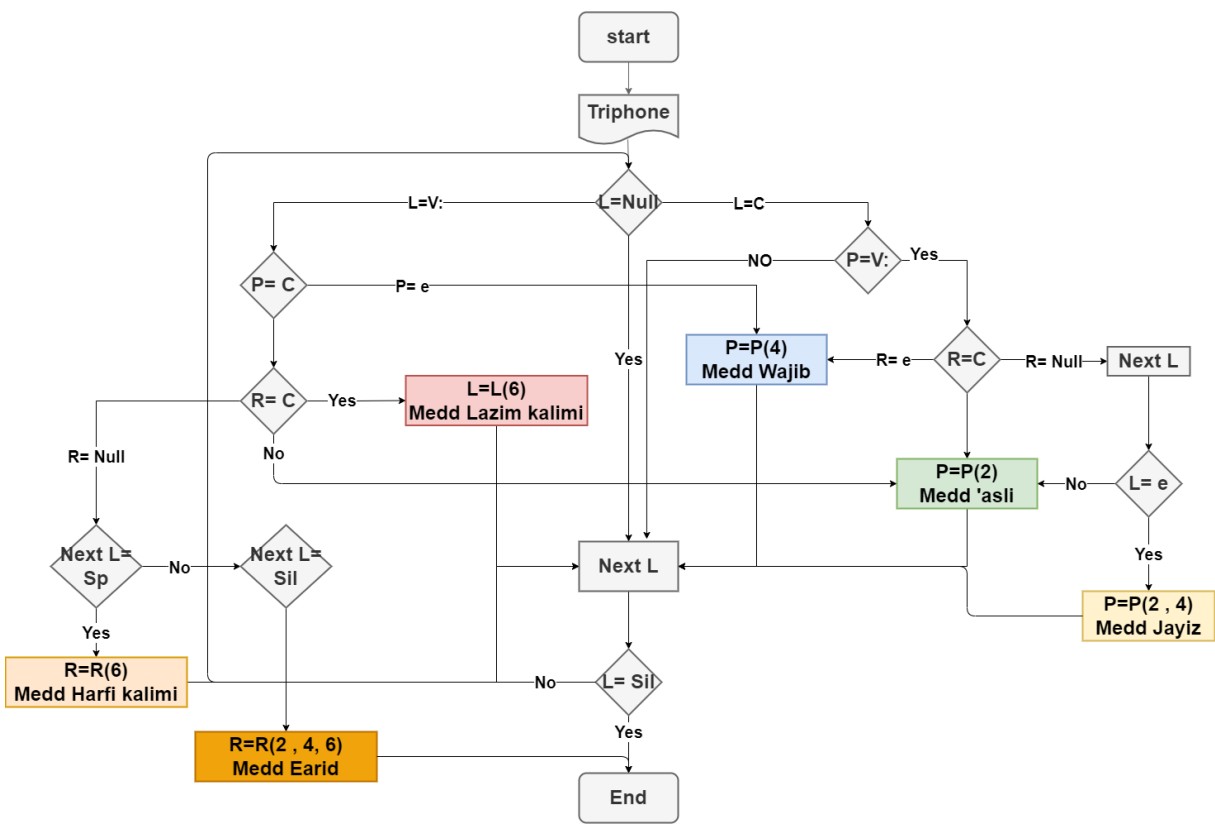

**Figure 13.** Identify Medd type using RPDAMC.

Notice that we faced many difficulties in writing our rule set as described above. The most important difficulty was the overlap of rules. This situation happens when more than one rule can be triggered for one letter. For instance, in the word "السماء" we have a lengthening of obligatory 4 Harakah, but when it is the last word in a verse, it changes to optionally 2, 4, or 6 Harakah. To overcome this overlap, we added to each rule a priority in order to avoid the conflict between triggered rules and by choosing only the rule with the highest priority.

Moreover, the Medd Earid LilSukoon is an optional Medd 2, 4, or 6 Harakah, but the reciter must adhere to what he chose, which means if he recites with 2, the rest of his recitation continues with that. Also, in the Qira'atu Hafs, there are two ways for the Medd Jayiz Munfasil, 2 or 4 Harakah. Although the most popular is that the letter lengthens by 4 Harakah, many of the imams of prayer chose the lengthening of 2 Harakah. In order to solve this problem, the first Harakah that the reciter reads will be saved as a required duration, as in the following:

D = 0;
For i = 1 To MaxP // MaxP is no. of last phoneme.
if (Medd_type = Jayiz Or Earid) // Medd type in CV-triphone then
Medd_Duration = D; // Medd_Duration the required duration.
if (duration = 2 Or 4 Or 6) // Duration can get from ANN-PDEM
if (D = 0) then
D = duration;
Medd_Duration = D;

### 3.3.4. Scientific Formulation of Medd Based on Tajweed Rules

To find a scientific formulation of the Medd rules in Tajweed, these rules were carefully studied by an expert in the Arabic language and the rules of Tajweed, where all possible and impossible cases of Medd in Qur'an recitation were investigated, and then all the

Medd rules described in the text by the scholars were deduced and written scientifically and thus are automatically processed by the computer.

The maximum number of words concerned with extracting the Tajweed rule is two; that is, either the previous word or the following word, and there are only six letters that are affected by the regulations, which are either preceding or following the letter [48].

With regard to the Medd rules, it relates to the long vowels; therefore, it will be considered a beginning and then looking at the letter before it and the letter that follows it, as well as the second letter after the Medd letter, which means the number of concerned letters is either three or four. Moreover, some types of Medd can be in one word, such as Medd Lazim, Medd Wajib, and Medd Earid LilSukoon. Other types can be in two words, such as Medd Jayiz. Also, the Medd 'asli can be in one or two words.

A whole scientific formulation has been proposed based on a new rule-based algorithm classification of Medd rules to be machine readable rules taking into consideration the following notations:

- "P" means the phoneme treated, "L" represents the phoneme preceding the phoneme treated, "R" represents the phoneme immediately following the phoneme treated, and "Next L" represents the first phoneme of the next word following the phoneme treated;
- "C" indicates consonants, "V" indicates short vowels, "V:" indicates long vowels, "e" indicates Hamza letter, and "*" is used as a wildcard character to replace any phoneme. These letters affect the rules of the Medd;
- "sp" represents a space character (between two words);
- The symbol "Sil" means silence;
- "Phoneme duration" means the required duration according to the type of Medd base in Tajweed rules;
- "D" indicates the chosen duration required for some types of Medd that depend on the choice of the reciter at the beginning of his recitation, such as Medd Asli Earid LilSukoon, where the reciter chooses to prolong the Medd letter by two, four, or six movements, and then he must adhere to what he chose;
- Finally, "Ruling" means the rule and type of Medd;

Since each type of Medd in the Qur'an has cases that can be identified by analysis of the Triphone tree, each case has a scientific formulation that will be detailed in the discussion of the results for each type of Medd.

Thus, as it is known in the Tajweed rules, Medd 'asli occurs if the Medd letter (long vowel) is not followed by a hamza letter or a letter has Sukoon or Shaddah. Therefore, 11 cases can be defined as the Medd 'asli, which are the cases described in Table 3. Table 3 also describes and explains the states of the rest of the Medd.

To describe the rules of Medd 'asli in a scientific formula that the computer understands, all cases of the Medd 'asli have been written in the following scientific formulas:

$$P = V: ; R = C; \text{Next L} <> C; \text{Ruling= Medd 'asli}$$

$$P = V: ; R = \text{blank}; \text{Next L} <> e; \text{Ruling = Medd 'asli}$$

$$L = V: ; P = C; R <> C; \text{Ruling = Medd 'asli}$$

Medd Wajib Mutasil occurs if the Medd letter (long vowel) is followed by a Hamza letter (symbolized in the Arabic Phonetic Dictionary by "e") in the same word. Therefore, the 9 cases can be defined as the Medd Wajib Mutasil, but these two scientific formulas describe this type:

$$P = V: ; R = e; \text{Next L} = *; \text{Ruling = Medd Wajib}$$

$$L = V: ; P = e; R <> C; \text{Ruling = Medd Wajib}$$

Medd Jayiz Munfasil occurs if the Medd letter (long vowel) is followed by a Hamza letter, where the Medd letter is the last in the word, and the Hamza letter is at the beginning of the next word. Therefore, two cases can be defined as the Medd Jayiz Munfasil, which can be described in one scientific formula:

P = V: ; R = blank; Next L = e; Ruling = Medd Jayiz Munfasil

Medd Lazim Kalimi occurs if the Medd letter (long vowel) is followed by permanent Sukoon or Letter has Shaddah. Therefore, two cases can be defined as the Medd Lazim Kalimi, which are the two cases described in this scientific formula:

L = V: ; P = C; R = C; Ruling = Medd Lazim Kalimi

Medd Lazim Harfi is occurring at the recitation of some letters that some Qur'an Surahs begin with. These letters are each individually read as a three-letter word, the middle letter being a Medd letter and the third letter having a permanent Sukoon. Therefore, one case can be defined as the Medd Lazim Harfi, which is the case described in the following formula:

L = V: ; P = C; R = C; Ruling = Medd Lazim Harfi

Medd Earid LilSukoon occurs when the reciter stops on one word with a presented Sukoon, and the last letter is preceded by the Medd letter. Therefore, one case can be defined as the Medd Earid LilSukoon, which is the case described in the following scientific formula:

L = V: ; P = C; R = blank; Next L = sil Ruling = Medd Earid LilSukoon

## 4. Result and Discussion

Through our proposed algorithm (RPDAMC), the HTK-generated form is developed by adding duration to the triphone tree. Figure 11. shows a sample of the HTK triphone tree, while Figure 14. shows the output of the rule-based duration phoneme algorithm for Medd classification.

**Figure 14.** Sample of triphone of RPDAMC.

It is noted that the developed triphone tree contains phonemes with the number of Harakah according to the Tajweed rules, in addition to defining the type of Medd, if any. The developed Triphone tree can be used in recognizing the duration of Medd or the duration of the phoneme in general, where the duration is matched with the number of required Harakah to find out whether the recitation is correct and follows the Tajweed rules.

This study conducted the statistics in the prompts sheet (that contains all selected Ayat), and the statistics were checked by an expert on the Tajweed rules (a teacher of the Holy Qur'an). Table 4 shows these statistics (real statistics). Statistics have shown the frequency of each type of Medd, which is as follows: Medd 'asli 26 times, Medd Wajib Mutasil 7 times, Medd Jayiz Munfasil 4 times, Medd Lazim Kalimi 5 times, Medd Lazim Harfi 3 times, Medd Earid LilSukoon 6 times. As well, Statistics of RPDAMC results were also performed and recorded. Table 4 shows these statistics.

**Table 4.** Real and RPDAMC results.

| Medd Type | Real Statistics | | Proposed Algorithm (RPDAMC) Results | | | |
|---|---|---|---|---|---|---|
| | TP | TN | TP | FP | TN | FN |
| Medd 'asli | 26 | 722 | 25 | 0 | 722 | 1 |
| Medd Wajib Mutasil | 7 | 741 | 7 | 0 | 741 | 0 |
| Medd Jayiz Munfasil | 4 | 744 | 4 | 0 | 744 | 0 |
| Medd Lazim Kalimi | 5 | 743 | 5 | 0 | 743 | 0 |
| Medd Lazim Harifi | 3 | 745 | 3 | 0 | 745 | 0 |
| Med Earid LilSukoon | 6 | 742 | 6 | 0 | 742 | 0 |

To measure precision, a confusion matrix has been used. A confusion matrix is "a table often used to define a classification model's performance on a set of test data for which the true values are known" [49].

The most well-known evaluation metric is accuracy which represents the ability to correctly differentiate the two cases [50]. The proportion of true positive and true negative should be calculated to estimate the accuracy of a test. Mathematically, accuracy is given by [51,52]:

$$\text{Accuracy (ACC)} = \frac{\text{TP} + \text{TN}}{\text{TP} + \text{TN} + \text{FP} + \text{FN}}$$

$$\text{Detection Rate or True Positive Rate (TPR)} = \frac{\text{TP}}{\text{TP} + \text{FN}},$$

$$\text{Precision (P)} = \frac{\text{TP}}{\text{TP} + \text{FP}},$$

$$\text{Recall or True Negative Rate(TNR)} = \frac{\text{TN}}{\text{TN} + \text{FP}},$$

$$\text{False Positive Rate (FPR)} = \frac{\text{FP}}{\text{TN} + \text{FP}}.$$

$$\text{False Negative Rate (FNR)} = \frac{\text{FN}}{\text{TP} + \text{FN}}.$$

where TP = true positive, TN= false negative, FN = false negative, FP = false positive.

A confusion matrix in Table 5 was used to measure the accuracy of the performance of the rule-based phoneme duration algorithm for Medd classification due to a confusion matrix that is often used to describe the performance of a classification model on a test dataset with known true values [53]. The four parameters, TP, FP, TN, and FN, were measured for each type of Medd separately.

**Table 5.** Confusion matrix rates for RPDAMC results.

| Medd Type | Confusion Matrix Rates % | | | | | | |
|---|---|---|---|---|---|---|---|
| | ACC | TPR | TNR | FPR | FNR | P | ERR |
| Medd 'asli | 99.87 | 96.51 | 100 | 0 | 3.85 | 100 | |
| Medd Wajib Mutasil | 100 | 100 | 100 | 0 | 0 | 100 | 0 |
| Medd Jayiz Munfasil | 100 | 100 | 100 | 0 | 0 | 100 | 0 |
| Medd Lazim Kalimi | 100 | 100 | 100 | 0 | 0 | 100 | 0 |
| Medd Lazim Harifi | 100 | 100 | 100 | 0 | 0 | 100 | 0 |
| Med Earid LilSukoon | 100 | 100 | 100 | 0 | 0 | 100 | 0 |

It should be noted that to obtain this high percentage of accuracy in performance, RPDAMC has gone through a series of improvements and procedures, including the following:

1.  The dictionary was checked manually, and errors in translating some words into phonemes were corrected, especially vowels and long vowels;
2.  Errors in the positions of Shaddah were observed in the dictionary, which led to a weakness in the accuracy of Medd Lazim Kalimi recognition. These errors have been corrected;
3.  Some Qur'an words were added to the dictionary, which are three letters representing the Medd Lazim Harfi: حم ,ق, and ن. This is because the automatic generation of the dictionary regards it as normal letters, but it is recited as words in the Qur'an recitation;
4.  Alif, which does not pronounce, was considered as Fathah in the dictionary. This means that the phoneme /aa:/ has been modified to /aa / and phoneme /ae:/ to /ae/ in all words that are not pronounced as Medd, such as "وَالْقُرْآنِ" it was like "w ae: l q ux r e ae: n ih sp" and modified to "w ae l q ux r e ae: n ih sp";
5.  The appropriate conditions and restrictions were added for each Medd type according to the cases.

## 5. Conclusions and Future Work

Qur'anic recitation is not like Arabic communication due to the rules that govern the recitation. The rules of duration are critical to determining the correct utterance and its meaning, while in Arabic communication, duration is not the main criterion to determine the correctness of the utterance; only word pattern is important. Thus, previous works which only enhanced speech recognition based on the pattern of the speech are not suitable for Qur'anic recitation. In fact, it is necessary to model the phoneme duration; therefore, this work introduced a rule-based algorithm embedded with Hidden Markov Model to solve this issue.

As the phoneme classification is the first step in phoneme recognition, the proposed algorithm classifies vowels in Qur'an recitation by their types based on the Tajweed rules and shows the duration of every type. First, methods for designing and pre-processing the dataset, including the language model, dictionary building, and creation of transcription, as well as creating a triphone tree, have been detailed. These steps are done by utilizing the HTK tool.

The main objective of this paper is to improve phoneme classification in Qur'anic recitation by developing an algorithm that has the ability to classify the types of Medd (vowels). The proposed algorithm used the triphone tree to classify the Medd according to its type in Tajweed rules. Therefore, all the accepted triphone cases in the Qur'an recitation were elaborated. The accepted cases were categorized into seven types of Medd based on the Tajweed rules; rules and restrictions were established to define each type of Medd separately. The results achieved the aim of the paper and showed that the proposed rule-based algorithm had improved the performance of phoneme classification according to Tajweed rules based on phoneme duration. RPDAMC results were compared

with the real statistics. The obtained performance results were high, and the accuracy of obtained classification results of Medd 'asli and the other six types was 99.87% and 100%, respectively.

The proposed algorithm will contribute significantly to Medd recognition. However, this work is an initial path to having more precise Qur'anic recitation learning tools, which can help many to learn and assist in Qur'anic recitation. For future work, there is a need to develop an algorithm to generate Arabic phonetic dictionaries for Qur'anic recitation recognition, as current Arabic dictionaries do not cover all issues of Qur'anic recitation, especially those rules of Tajweed that are distinct for Qur'anic recitation from Arabic speech, as well as the differences in some writing rules. Moreover, although the Tajweed rule is fixed, the duration as the way of Qur'anic recitation varies and thus requires more data for training.

**Author Contributions:** Conceptualization, A.M.A.A., M.S.S. and M.S.H.S.; methodology, A.M.A.A. software, A.M.A.A. formal analysis, A.M.A.A. and R.A.; investigation, A.M.A.A.; data curation, A.M.A.A.; writing original draft preparation, A.M.A.A.; writing review and editing, M.S.S., M.S.H.S., A.A.S.A., S.T., M.A.H.A. and A.A.S.; visualization, A.M.A.A. and A.A.S.; supervision, M.S.S., M.S.H.S. and R.A.; funding acquisition, R.A., S.T. and M.A.H.A. All authors have read and agreed to the published version of the manuscript.

**Funding:** This research was funded by the United Arab Emirates UAEU-ZU Joint Research Grant G00003715 (Fund No.: 12T034) through Emirates Center for Mobility Research.

**Acknowledgments:** The authors would like to thank the United Arab Emirates University for funding this work under UAEU-ZU Joint Research Grant G00003715 (Fund No.: 12T034) through Emirates Center for Mobility Research. Also, the authors would like to thank the Research Management Center, Malaysia International Islamic University, for funding this work with Grant RMCG20-023-0023. We also would like to thank the Universiti Teknologi Malaysia for supporting and providing the opportunity to conduct this research work.

**Conflicts of Interest:** The authors declare that there are no known conflict of interest associated with this publication, and there has been no significant financial support for this work that could have influenced its outcome.

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
