# Peer review of "Rule-Based Embedded HMMs Phoneme Classification to Improve Qur’anic Recitation Recognition"

_electronics, doi:10.3390/electronics12010176_

Round 1
Reviewer 1 Report
This paper has a high thematic relevance since it carries out an exhaustive analysis. However some recommendations must be followed:
- the introduction section must present the objectives of the study, a major finding, and the structure of the article.
- between the following sub-sections: 3.1. and 3.1.1. must include a transition paragraph; for 3.3 and 3.3.1. as well;
For the final section, number 5, you must address better the aim of the study, the objectives (if they were achieved or not), and the implications (theoretical and managerial).
Also, some references must be revised due to the setup; I have observed some errors.
I wish you good work!
Author Response
The authors would like to thank the Reviewer for his comments. The reviewer’s comments were taken into account and all the changes were marked in yellow in the revised manuscript.
Kindly see the attachments

Reviewer 2 Report
This paper targets phoneme classification algorithms, especially for the Islamic language, which is very practical and interesting. The Rule-Based Phoneme Duration Algorithm to improve phoneme classification is proposed and tested with high accuracy. The overall quality of this research is good. I have no further comments on this manuscript.
Author Response
The authors would like to thank the Reviewer for his comments.
Reviewer 3 Report
This article presents a rule-based phoneme duration algorithm to improve phoneme classification in Quranic recitation. The proposed algorithm achieved outstanding accuracy, ranging from 99.87% to 100% depending on the Medd type. The results obtained from the proposed algorithm will contribute significantly to the recognition models of Qur'anic recitations.
The study is well planned. The base is adequate, the methodological design successful. The results come close to effectively reflecting the findings of the study. The conclusions respond to the proposed objectives and with current references. Its publication is recommended given the aforementioned conditions and the interest of the topic. For all the above, I consider that this work can be published.
Thank you
Author Response

(The authors gave the same response as above.)

Round 2
Reviewer 1 Report
Good work!